# Silkworm model of biofilm formation: *In vivo* evaluation of antimicrobial tolerance of a cross-kingdom dual-species (*Escherichia coli* and *Candida albicans*) biofilm on catheter material

**Shintaro Eshima, Yasuhiko Matsumoto⦿\*, Sanae Kurakado, Takashi Sugita**

Department of Microbiology, Meiji Pharmaceutical University, Kiyose, Tokyo, Japan

\* ymatsumoto@my-pharm.ac.jp

## Abstract

Biofilms are formed by microorganisms and their products on the surface of materials such as medical devices. Biofilm formation protects microorganisms from antimicrobial agents. Bacteria and fungi often form dual-species biofilms on the surfaces of medical devices in clinical settings. An experimental system to evaluate *in vivo* biofilm formation by the patho-genic fungus *Candida albicans* was established using silkworms inserted with polyurethane fiber (PF), a catheter material. In the present study, we established an *in vivo* experimental system using silkworms to evaluate the antimicrobial tolerance of *Escherichia coli* in single- and dual-species biofilms formed on the surface of the PF. The injection of *E. coli* into the PF-inserted silkworms led to the formation of a biofilm by *E. coli* on the surface of the PF. *E. coli* in the biofilm exhibited tolerance to meropenem (MEPM). Furthermore, when *E. coli* and *C. albicans* were co-inoculated into the PF-inserted silkworms, a dual-species biofilm formed on the surface of the PF. *E. coli* in the dual-species biofilm with *C. albicans* was more tolerant to MEPM than *E. coli* in the single-species biofilm. These findings suggest the usefulness of an *in vivo* experimental system using PF-inserted silkworms to investigate the mechanisms of MEPM tolerance in *E. coli* in single- and dual-species biofilms.

## Introduction

Biofilms are 3-dimensional structures comprising microorganisms and an extracellular matrix of polysaccharides, proteins, nucleic acids, and lipids [1, 2]. Microorganisms form biofilms on the surfaces of medical devices [3, 4]. Biofilm formation confers tolerance to antimicrobial agents, leading to recurrent and chronic infections [3, 5, 6]. Medical treatment of biofilm infec-tions, such as prosthetic joint infections and infective endocarditis after valve replacement sur-gery, is challenging even with antimicrobial agents that are sensitive *in vitro* [7, 8]. Biofilm-associated infections, which account for 65% of nosocomial infections, impose a significant burden on healthcare, including the surgical removal of medical devices with biofilm forma-tion and the administration of multiple antimicrobial agents [3, 9].

**Funding:** This study was supported in part by the Research Program on Emerging and Re-Emerging Infectious Diseases of the Japan Agency for Medical Research and Development (grant number JP23fk0108679h0401 to TS) and for JSPS KAKENHI grant number JP23K06141 (Scientific Research (C) to Y.M.). The funders had no role in the study design, data collection, data analysis, decision to publish, or preparation of the manuscript. There was no additional external funding received for this study.

**Competing interests:** The authors have declared that no competing interests exist.

Pathogenic microorganisms form a multi-species biofilm that causes 15% of catheter-related bloodstream infections (CRBSI), which are complex infections [10]. The incidence of bloodstream infections (BSI) due to *Escherichia coli* has increased worldwide over the past decade, with mortality rates ranging from 5% to 30% [11–13]. Overcoming biofilm-associated infections by *E. coli* is an important issue because approximately 22% of BSIs are CRBSI [14]. *Candida albicans* is the most frequently isolated fungus from the blood and forms cross-kingdom biofilms with pathogenic bacteria [15, 16]. Meropenem (MEPM) tolerance in *E. coli* is induced by the formation of a dual-species biofilm with *C. albicans in vitro* [17]. Therefore, a dual-species biofilm of *E. coli* and *C. albicans* may complicate treatment with antimicrobial agents against *E. coli* in the clinical setting.

Biofilm formation by pathogenic microorganisms in infected hosts is influenced by the host cells, proteins, and nutrients [18, 19]. Therefore, *in vivo* experimental systems are important for evaluating biofilm formation by pathogenic microorganisms [19]. Techniques to evaluate biofilm formation by *C. albicans* on the surfaces of jugular vein catheters have been established in mammalian animals, such as mice, rats, and rabbits [20–23]. Experimental systems for evaluating *in vivo* biofilm formation by pathogenic microorganisms require large numbers of animals, however, and the use of mammals is associated with higher housing and maintenance costs, more complicated procedures, and ethical issues.

Silkworms have several advantages over mammals as experimental animals, including low rearing costs, the ability to rear large numbers of animals in a small space, and fewer ethical issues associated with their use in research [24]. The silkworm is a thus useful experimental animal model for evaluating the virulence of pathogenic microorganisms in systemic infections and the efficacy of antimicrobial agents [25, 26]. Quantitative injection of a sample solution and harvesting of the hemolymph in silkworms are simpler than the equivalent procedures in mammals [27–29]. The doses of antimicrobial drugs per body weight required to treat infected silkworms and mammals are comparable [30]. Further, novel antimicrobial compounds found to be therapeutically effective in mouse infection models were identified by *in vivo* screening using silkworm infection models [31, 32]. While an experimental infection system using medical device substrate-inserted silkworms was established for *in vivo* evaluation of biofilms caused by *C. albicans* [33], an experimental system using silkworms that enables the evaluation of *E. coli* biofilms *in vivo*, has not yet been developed.

In this study, we established an experimental system for evaluating biofilm formation by *E. coli* on the surface of a polyurethane fiber (PF) inserted in a silkworm and investigated the effect of MEPM against a dual-species biofilm with *C. albicans*. Polyurethane is a central venous catheter material [34]. *E. coli* in a single-species biofilm on the surface of the PF in the silkworm was tolerant to MEPM. Formation of a dual-species biofilm with *E. coli* and *C. albicans* on the surface of the PF in the silkworm further enhanced the MEPM tolerance of *E. coli* in the biofilms. Our findings suggest that the PF-inserted silkworm infection model is useful for evaluating MEPM tolerance in *E. coli*.

## Materials & methods

### Reagents

Meropenem was purchased from FUJIFILM Wako Pure Chemical Corporation (Osaka, Japan) and dissolved in physiologic saline (0.9% NaCl). Micafungin was purchased from Sigma-Aldrich, Inc. (St. Louis, MO, USA), dissolved in distilled water, and stored at −80˚C until use.

## Bacterial and fungal strains

*E. coli* RB-3 and *C. albicans* SC5314 strains were used in this study. The *E. coli* RB-3 strain was obtained from blood cultures of patients with urinary tract infections treated at Toshiba Rinkan Hospital, Kanagawa, Japan, in 2014 [17]. *E. coli* was grown on a nutrient agar medium and incubated at 37˚C for 24 h. *C. albicans* SC5314 was grown on Sabouraud dextrose agar at 27˚C for 24 h.

## Silkworm rearing

Silkworms were reared as previously described (34). Silkworm eggs (Hu Yo × Tukuba Ne) were purchased from Ehime-Sanshu Co., Ltd. (Ehime, Japan), disinfected, and hatched at 25˚C–27˚C. Silkworms were fed an artificial diet, Silkmate 2S (Ehime-Sanshu Co., Ltd., Ehime, Japan) mixed with vancomycin (300 μg/g of Silkmate 2S). Fifth-instar silkworms were fed with an antibiotic-free artificial diet (Sysmex Corporation, Hyogo, Japan) for 1 day.

## Polyurethane fiber-inserted silkworms

A PF was inserted into the silkworms as described previously [33]. The PF (thickness: 0.5 mm, Gomutegusu F046, No. H3; Daiso-Sangyo, Hiroshima, Japan) was cut into 2-cm lengths, treated with a 70% ethanol solution for 15 min, and then dried under UV irradiation for 30 min. A hole was punctured on the back of each silkworm using a marking pin (Daiso-Sangyo), and a UV-sterilized PF was then inserted into the silkworm body through the hole. The PF-inserted silkworms were observed at room temperature for 30 min to ensure that the bleeding stopped.

## Infection experiments using silkworms

Silkworm infection experiments were performed as previously described [35]. Fifth-instar larvae were fed an artificial diet (Silkmate 2S; Ehime-Sanshu Co. Ltd., Ehime, Japan) overnight. A cell suspension (50 μL) was injected into the silkworm hemolymph using a 1-ml tuberculin syringe (Terumo Medical Corporation, Tokyo, Japan). After incubation at 27˚C for 18 h, a drug solution (50 μL) was injected into the silkworms and the PFs were recovered from the silkworms at 1 h after the drug injection.

## Crystal violet staining

Crystal violet staining of the biofilm on the surface of the PF was performed according to a previously described method [33] with slight modifications. The PFs recovered from the silkworms were transferred to a 1.5-ml tube, washed twice with saline, and treated with methanol for 20 min. After removing the methanol solution, the PFs were dried for 1 h. A 0.1% (w/v) aqueous crystal violet solution (350 μL) was added to the tube and incubated at room temperature for 20 min. After removing the staining solution, the PFs were washed twice with 20% ethanol and once with distilled water. The biofilm on the PF surface was observed under a microscope (CH-30; Olympus, Tokyo, Japan). After microscopic observation, the PFs were placed in 33% (v/v) acetic acid (500 μL) for 30 min and distilled water (500 μL) was added. The absorbance (at A590) of each solution was measured.

## Viable cell counts in biofilms on PF

Viable cells in biofilms were determined by a colony-forming unit (CFU) assay according to a previous report [17]. Saline (500 μL) was added to a 1.5-mL tube containing PFs recovered from silkworms, and biofilm cells were resuspended by vortexing for 15 min. Dilution series

were prepared and plated on nutrient agar. After incubation, colonies were counted and CFUs were calculated. The CFUs of *E. coli* and *C. albicans* were calculated: 1 μg/mL micafungin was added to nutrient agar medium (to selectively grow only *E. coli*) or 100 μg/mL streptomycin was added to Sabouraud agar medium (to selectively grow only *C. albicans*). *E. coli* RB-3 and *C. albicans* SC5314 strains were susceptible to streptomycin and micafungin, respectively.

### Statistical analysis

The significance of the difference between 2 groups was calculated using Student's *t*-test. Statistical significance in the dose-dependence experiments was determined using Dunnett's test. The significance of differences between multiple groups was assessed using the Tukey-Kramer method. Statistical significance was set at $P < 0.05$.

## Results

### Biofilm formation by *E. coli* on PF in the silkworms

We determined the cell number of *E. coli* required for biofilm formation on the surface of the PFs in silkworms. The PFs were isolated from the PF-inserted silkworms 18 h after injection of *E. coli* cells and stained with crystal violet (Fig 1A). Absorbance at 590 nm ($A_{590}$) of crystal violet dissolved from PFs in the group of silkworms inoculated with *E. coli* ($2 \times 10^8$ cells/larva) was 3-fold higher than that in the saline-inoculated group (Fig 1B and 1C). Moreover, the viable cell number of *E. coli* on PFs isolated from the silkworms increased in an injected dose-dependent manner (Fig 2).

### Effect of MEPM on *E. coli* biofilm in the PF-inserted silkworms

Next, we examined the effect of MEPM on *E. coli* that formed a biofilm in the silkworms. MEPM reduces the number of viable *E. coli* cells in the biofilm by treatment with 12.5 and 50 μg/mL *in vitro* [17]. Because the volume of silkworm hemolymph used in this study was approximately 500 μL, administration of 3.1–50 μg MEPM into silkworms reaches 6.2–100 μg/mL in the silkworm hemolymph, which is comparable to the effective concentration *in vitro*. The *E. coli* cells ($2 \times 10^8$ cells/larva) were inoculated into PF-inserted silkworms. At 18 h after inoculation, MEPM solution (0–50 μg/larva) was administered and the silkworms were incubated for 1 h (Fig 3A). The number of viable *E. coli* cells on the PF surface in silkworms administered MEPM (0–50 μg/larva) was not altered compared with the number in silkworms administered only saline (Fig 3B). We next examined whether administering a higher dose of MEPM reduced the number of viable *E. coli* cells that formed a biofilm in the silkworms. A high-dose MEPM solution (0–1000 μg/larva) was administered to PF-inserted silkworms inoculated with *E. coli* cells ($2 \times 10^8$ cells/larva) and the silkworms were incubated for 1 h (Fig 3A). The number of viable *E. coli* cells on the PF surface was decreased in the silkworms administered MEPM at 62.5, 250, or 1000 μg/larva compared with silkworms administered saline (Fig 3C). The effect of MEPM, however, did not significantly differ among the 62.5, 250, and 1000 μg/larva doses. These results suggest that administering 62.5 μg MEPM per larva effectively decreased the number of viable *E. coli* cells in the biofilm that formed on the PFs in the silkworms.

### Dual-species biofilm formation by *E. coli* and *C. albicans* in the silkworms

The *E. coli* RB-3 and the *C. albicans* SC5314 strains form a dual-species biofilm *in vitro* [17]. We examined whether *E. coli* and *C. albicans* form a dual-species biofilm in PF-inserted silkworms. The surface of the PFs removed from the PF-inserted silkworm bodies at 18 h after

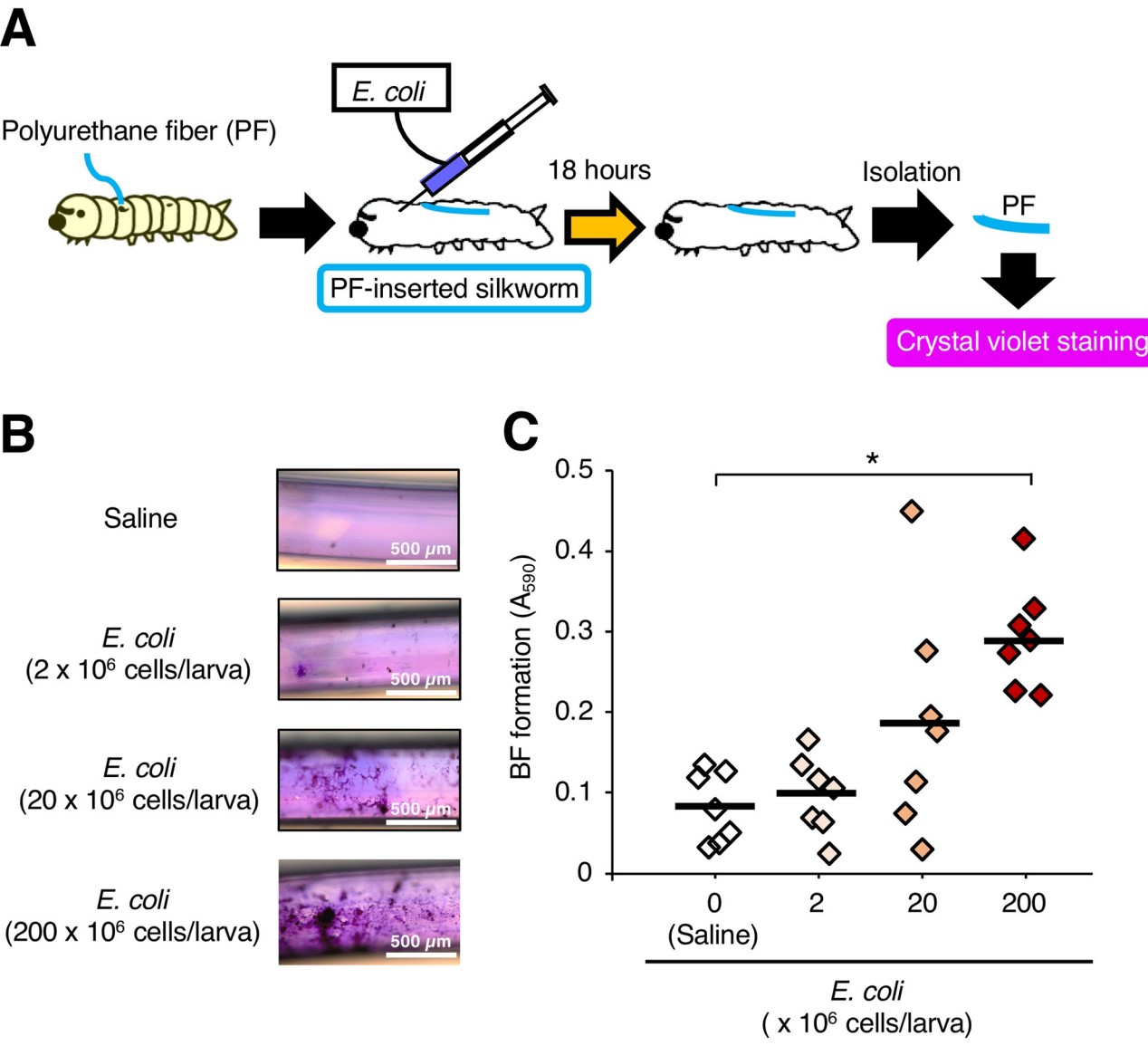

**Fig 1. Biofilm-formation by *E. coli* on the surface of the PFs in the silkworms.** (**A**) Experimental scheme. Saline or *E. coli* cell suspension (2 x $10^6$–2 x $10^8$ cells/50 μL) was inoculated into PF-inserted silkworms, and biofilms on PFs isolated after 18 h of rearing at 27°C were stained with crystal violet. (**B**) Microscopic observation. (**C**) The biofilm mass was determined by measuring absorbance 590 nm ($A_{590}$). n = 7/group. Statistically significant differences between groups were evaluated using Tukey's test. * $P < 0.05$.

inoculation with *E. coli* and/or *C. albicans* cells was stained with crystal violet (Fig 4A). Absorbance at 590 nm ($A_{590}$) of the crystal violet dissolved from the PFs in the group of silkworms inoculated with *E. coli* and *C. albicans* was higher than that in silkworms inoculated with *E. coli* only (Fig 4C). The viable cell number of *E. coli* and *C. albicans* on the PFs removed from the silkworms was not decreased by their co-inoculation (Fig 5).

### Effect of MEPM against the dual-species biofilm formed by *E. coli* and *C. albicans* in silkworms

The *C. albicans* SC5314 strain promotes MEPM tolerance of the *E. coli* RB-3 strain in a dual-species biofilm *in vitro* [17]. We therefore examined whether *E. coli* exhibits MEPM tolerance

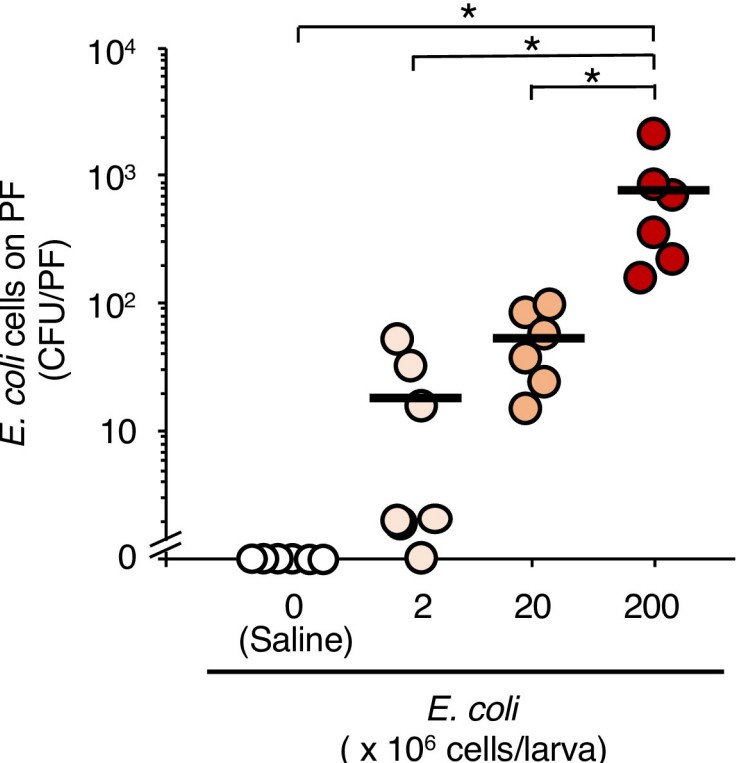

**Fig 2. Viable *E. coli* cells in the biofilm on the surface of the PFs in the silkworms.** Saline or *E. coli* cell suspension (2 x 10^6–2 x 10^8 cells/50 μL) was inoculated into PF-inserted silkworms, and the viable *E. coli* cells in the biofilms on PFs isolated after 18 h of rearing at 27˚C were determined by the CFU method. n = 7/group. Statistically significant differences between groups were evaluated using Dunnett's test. * $P < 0.05$.

by forming a dual-species biofilm with *C. albicans* in the silkworms (Fig 6A). The number of viable *E. coli* cells on the surface of the PFs in the silkworms administered MEPM (250 μg/ larva) was increased by co-inoculation of *C. albicans* (1 x 10^6 cells/larva) (Fig 6B). On the other hand, co-inoculation of *C. albicans* at 10^5 cells/larva with *E. coli* did not increase the number of viable *E. coli* cells in silkworms injected with MEPM (Fig 6C).

### Combination of MEPM and micafungin against a dual-species biofilm formed by *E. coli* and *C. albicans*

We investigated whether the co-administration of micafungin (MCFG), an antifungal drug, with MEPM affects the cell viability of *E. coli* in a dual-species biofilm in silkworms. A solution of *E. coli* cells (2 x 10^8 cells/larva) and *C. albicans* cells (1 x 10^6 cells/larva) was inoculated into PF-inserted silkworms. At 18 h after injection, a solution of MEPM (250 μg/larva) with or without MCFG (10 μg/larva) was administered and the silkworms were incubated for 1 h. The number of viable *E. coli* and *C. albicans* cells on the surface of the PFs in silkworms administered MEPM and MCFG was not altered compared to that in silkworms administered MEPM only (Fig 7).

### Discussion

In the present study, we established an *in vivo* experimental system of biofilm formation by *E. coli* in silkworms to evaluate the tolerance of *E. coli* to MEPM and the enhanced MEPM

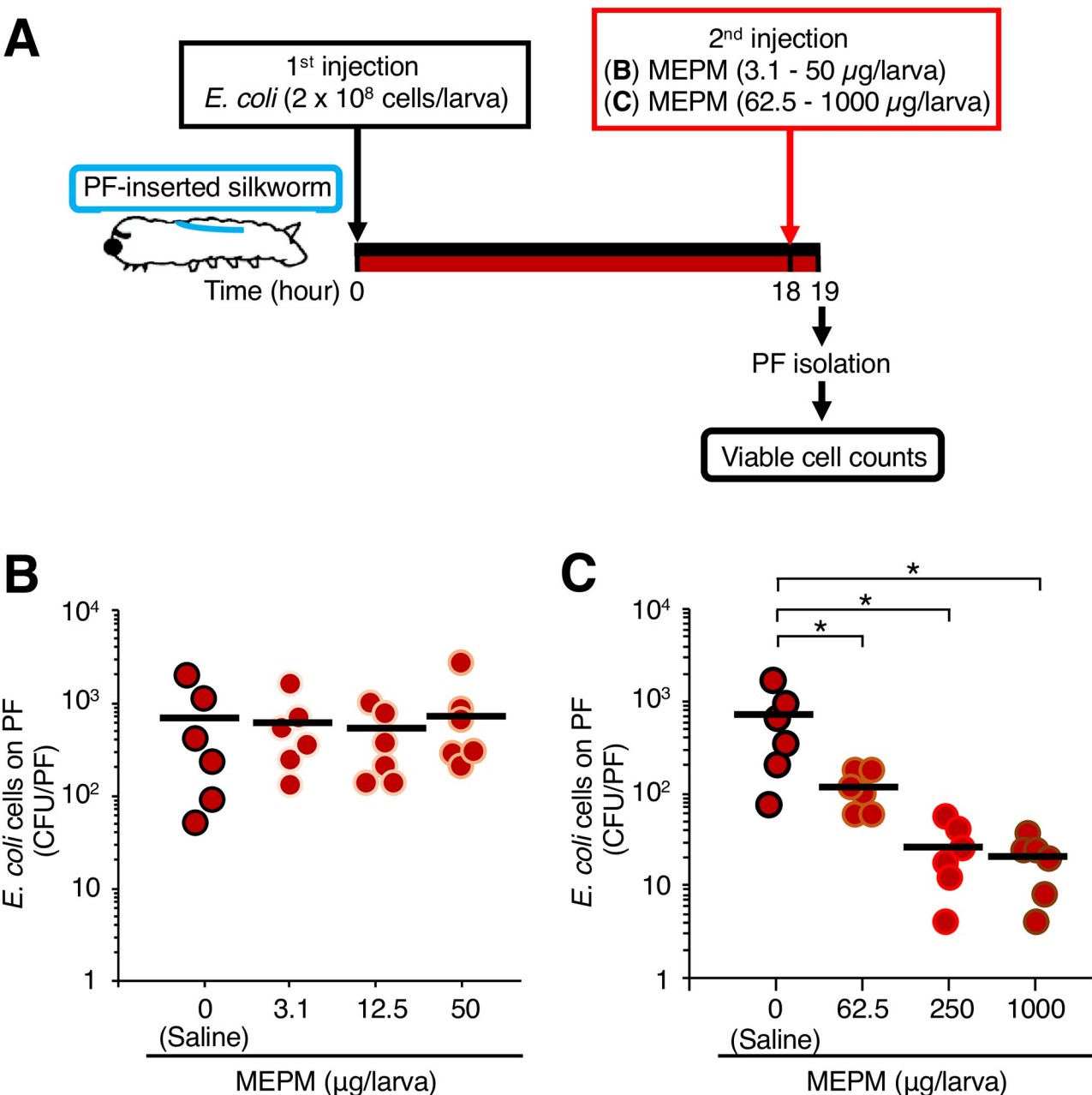

**Fig 3. Effect of MEPM on viable *E. coli* cells forming a biofilm on the surface of the PFs in the silkworms.** (**A**) Experimental scheme. *E. coli* cell suspension (2 x 10$^8$ cells/50 μL) was inoculated into PF-inserted silkworms, and the infected silkworms were incubated at 27°C for 18 h. After incubation, saline or MEPM solution (3.1–1000 μg/50 μL) was administered, and the silkworms were incubated at 27°C for 1 h. (**B, C**) Viable *E. coli* cells on the surface of the PFs in the silkworms administered MEPM (3.1–50 μg/50 μL) (**B**) or MEPM (62.5–1000 μg/50 μL) (**C**) were measured. n = 6/ group. Statistically significant differences between groups were evaluated using Dunnett's test. $^*$ $P < 0.05$.

tolerance of *E. coli* by *C. albicans*. The *in vivo* system allowed for successful evaluation of biofilm formation by *E. coli* and thus has potential use in studies investigating antimicrobial tolerance by biofilm formation.

We determined the conditions in which the clinical isolate *E. coli* RB-3 strain forms biofilms on the surface of the PFs in silkworms. Moreover, an experimental system was developed

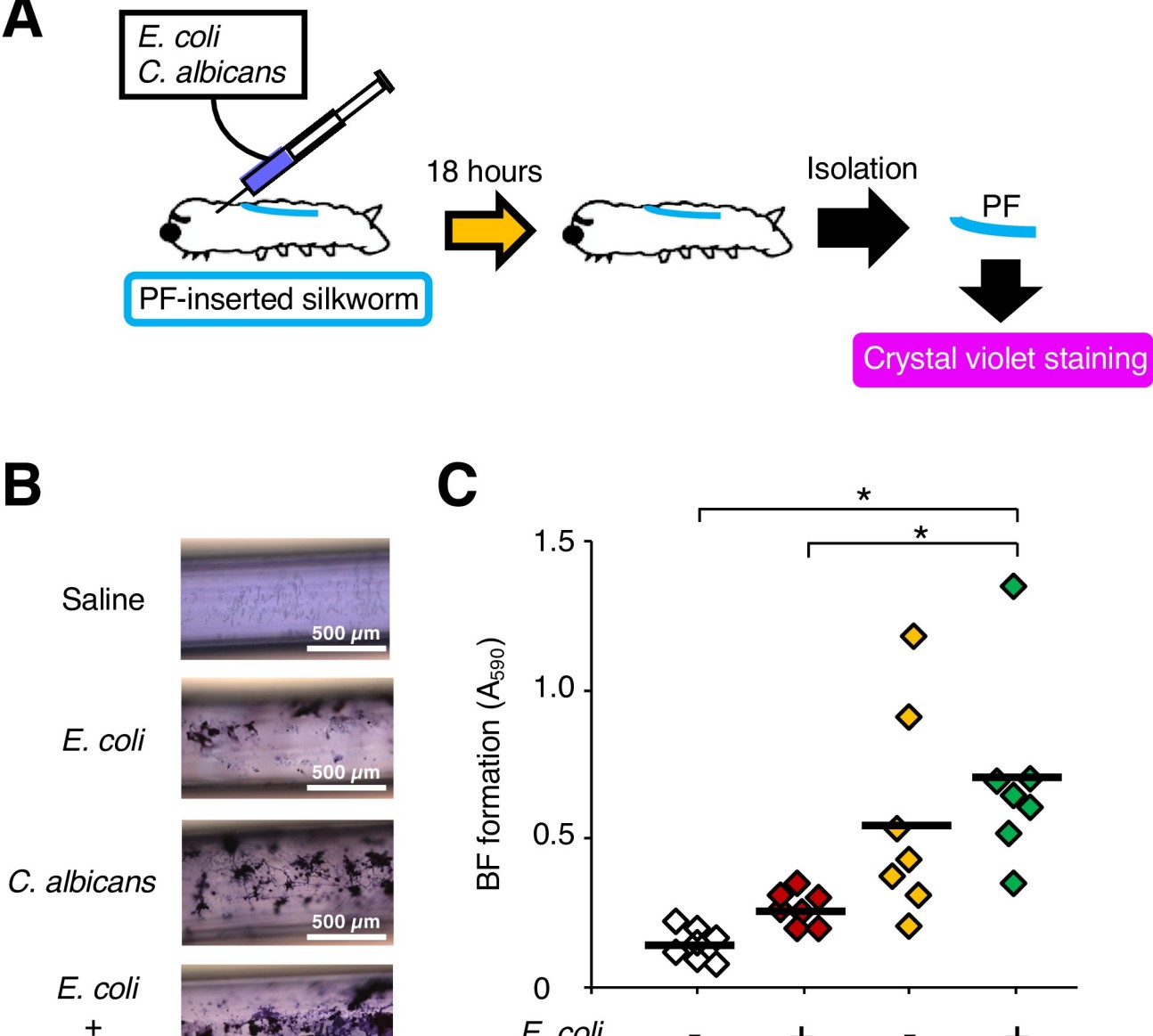

**Fig 4. Biofilm formation by *E. coli* and *C. albicans* on the surface of the PFs in the silkworms.** (**A**) Experimental scheme. Saline, *E. coli* cell suspension ($2 \times 10^8$ cells/50 μL), *C. albicans* cell suspension ($10^6$ cells/50 μL), or mixed cell suspension (*E. coli*: $2 \times 10^8$ cells and *C. albicans*: $10^6$ cells/50 μL) was inoculated into PF-inserted silkworms, and biofilms on PFs isolated after 18 h of rearing at 27°C were stained with crystal violet. (**B**) Microscopic observation. (**C**) The biofilm mass was determined by measuring absorbance 590 nm ($A_{590}$). n = 7/group. Statistically significant differences between groups were evaluated using Tukey's test. * $P < 0.05$.

to count the number of viable *E. coli* cells in the biofilm that forms on the surface of the PFs in the silkworms (S1 Fig in S1 File). An experimental system for biofilm formation by *C. albicans* on the surface of the PF in silkworms was established, but the conditions for counting the numbers of viable *C. albicans* cells in the biofilm forming on the PF surface in silkworms have not been established [33]. *C. albicans* forms a biofilm on the surface of the PFs in silkworms by hyphal elongation [33]. Biofilm formation by *C. albicans* on the PF surface in silkworms was

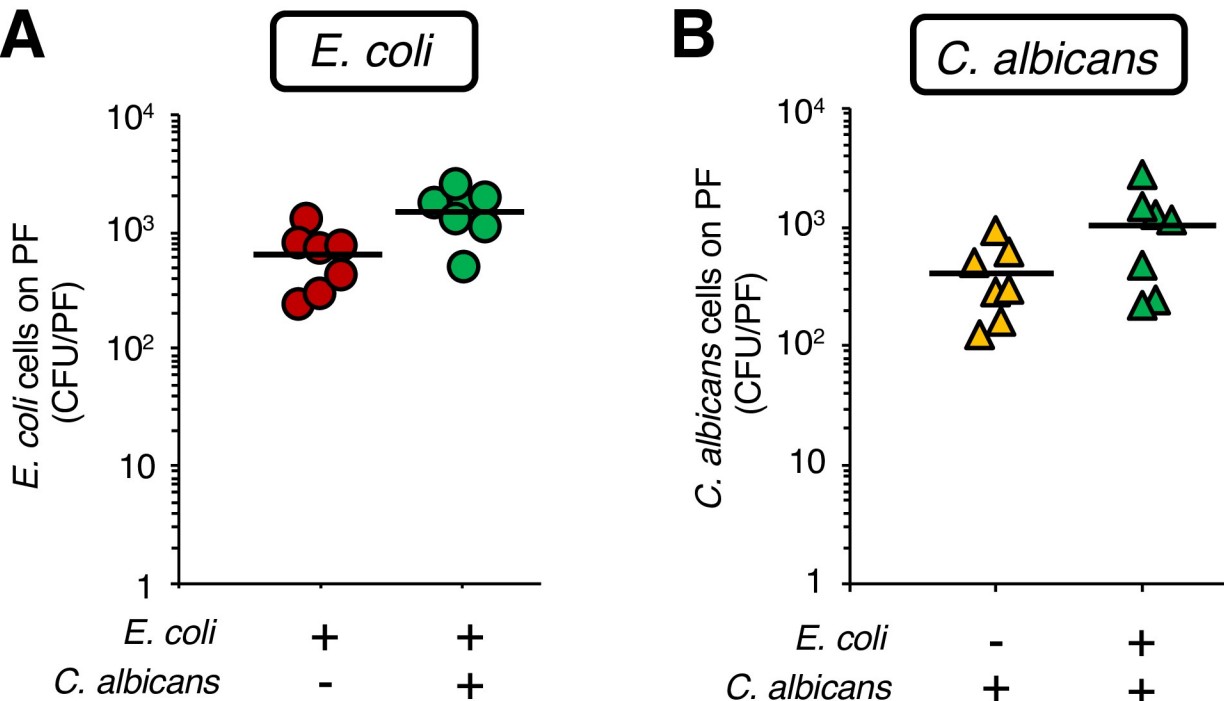

**Fig 5. Viable *E. coli* and *C. albicans* cells in the dual-species biofilm on the surface of the PFs in the silkworms.** (**A**, **B**) *E. coli* cell suspension (2 x 10$^8$ cells/50 μL), *C. albicans* cell suspension (1 x 10$^6$ cells/50 μL), or a mixed cell suspension (*E. coli*: 2 x 10$^8$ cells and *C. albicans*: 1 x 10$^6$ cells/ 50 μL) were inoculated into PF-inserted silkworms, and the viable *E. coli* cells (**A**) and *C. albicans* cells (**B**) in the biofilms on PFs isolated after 18 h of rearing at 27°C were determined by the CFU method. n = 7/group. Statistically significant differences between groups were evaluated using Student's *t*-test. *$P < 0.05$.

determined by crystal violet staining in a previous report [33]. The present study is the first to establish an experimental system to evaluate biofilm formation by counting the number of viable bacterial cells on the surface of PFs in silkworms. In this study, we investigated biofilm formation at 18 h after inoculation with *E. coli* and *C. albicans*. In an *in vitro* experiment, *E. coli* inhibits the growth of *C. albicans* [36]. We assumed that prolonged co-infection in the host environment was necessary for the formation of a dual-species biofilm for tolerance to MEPM. Establishing an experimental system using silkworms to evaluate various bacteria that form biofilms on catheter surfaces is a topic for future studies.

Silkworms are useful animals for evaluating the toxicity and efficacy of therapeutic agents for humans [24]. Silkworms and mammals exhibit comparable compound toxicities per body weight [37]. Because the body weight of silkworms is lower than that of mice and rats, compound toxicity can be evaluated with smaller doses [37]. Moreover, doses per body weight of antimicrobial drugs have similar therapeutic efficacies between silkworms and mammals [30, 31]. For the treatment of patients with sepsis and septic shock, MEPM is administered at 3–6 g (50–100 μg/g weight) per day [38]. Assuming a human adult body weight of 60 kg, the administration doses are 50–100 μg per 1 g of body weight (50–100 μg/g weight). Because the body weight of silkworms used in this study was approximately 2 g, the administration of 100 μg of MEPM into silkworms was comparable to a daily dose in humans (50 μg/g weight). The number of viable *E. coli* cells in the biofilm that formed on the PF surface in the silkworms decreased when 250 μg of MEPM was injected into the silkworms, but surviving *E. coli* remained. Therefore, the high dose of MEPM is not sufficient to eliminate all *E. coli* in the biofilm that formed on the surface of the PFs in the silkworms. The *E. coli*

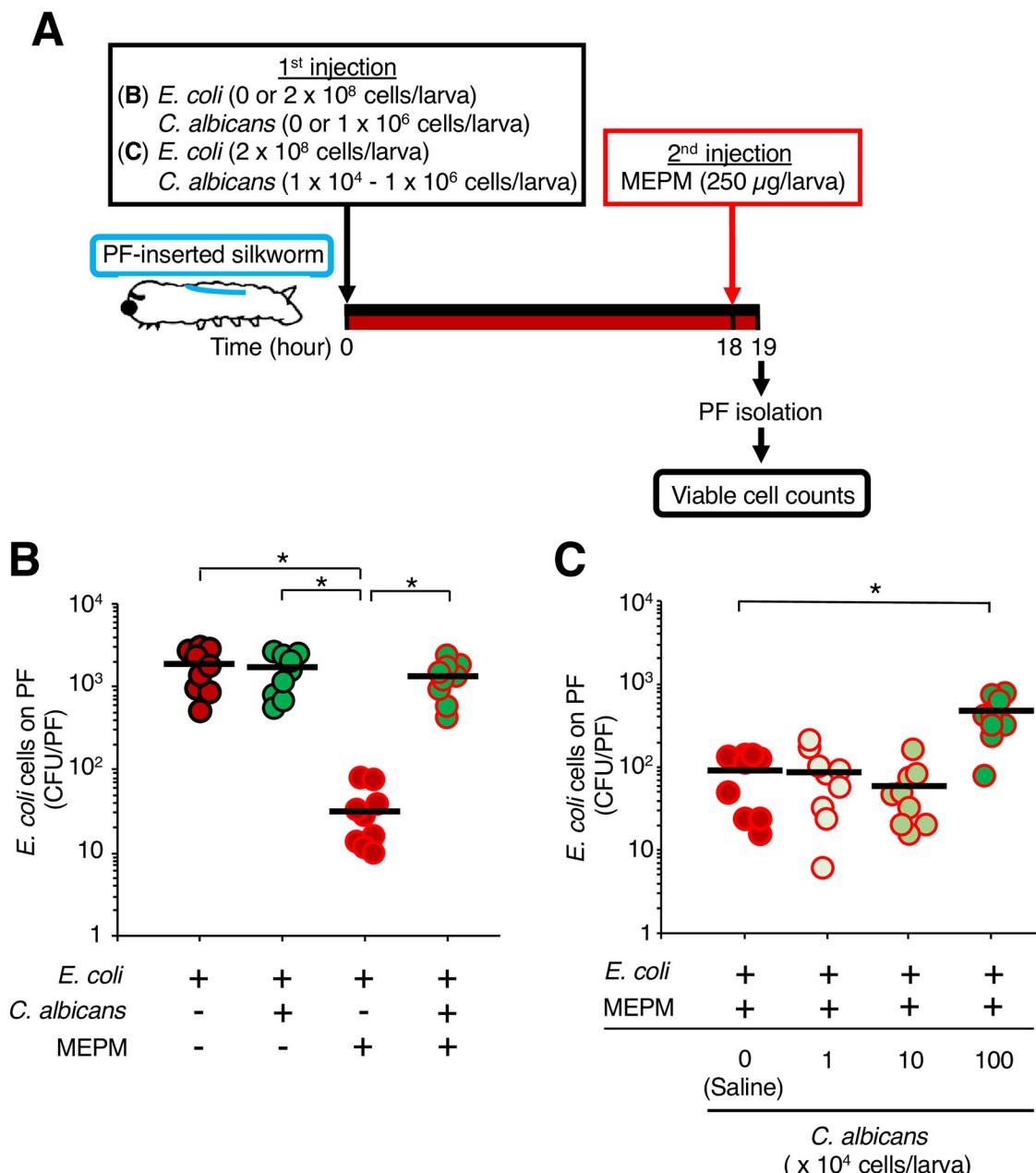

**Fig 6. MEPM tolerance of *E. coli* induced by *C. albicans* in a dual-species biofilm on the surface of the PFs in silkworms.** (A) Experimental scheme. (**B**) *E. coli* cell suspension (2 x 10^8 cells/50 μL) or a mixed cell suspension (*E. coli*: 2 x 10^8 cells and *C. albicans*: 1 x 10^6 cells/50 μL) was inoculated into PF-inserted silkworms, and the infected silkworms were incubated at 27°C for 18 h. After incubation, saline or MEPM solution (0 or 250 μg/50 μL) was administered, and the silkworms were incubated at 27°C for 1 h. Viable *E. coli* cells on the surface of the PFs in the silkworms were measured. n = 9/group. Statistically significant differences between groups were evaluated using Tukey's test. * $P < 0.05$. (**C**) *E. coli* cell suspension (2 x 10^8 cells/50 μL) or mixed cell suspension (*E. coli*: 2 x 10^8 cells and *C. albicans*: 10^4–10^6 cells/50 μL) was inoculated into PF-inserted silkworms, and the infected silkworms were incubated at 27°C for 18 h. After incubation, saline or MEPM solution (250 μg/50 μL) was administered, and the silkworms were incubated at 27°C for 1 h. Viable *E. coli* cells on the surface of the PFs in the silkworms were measured. n = 9/group. Statistically significant differences between groups were evaluated using Dunnett's test. * $P < 0.05$.

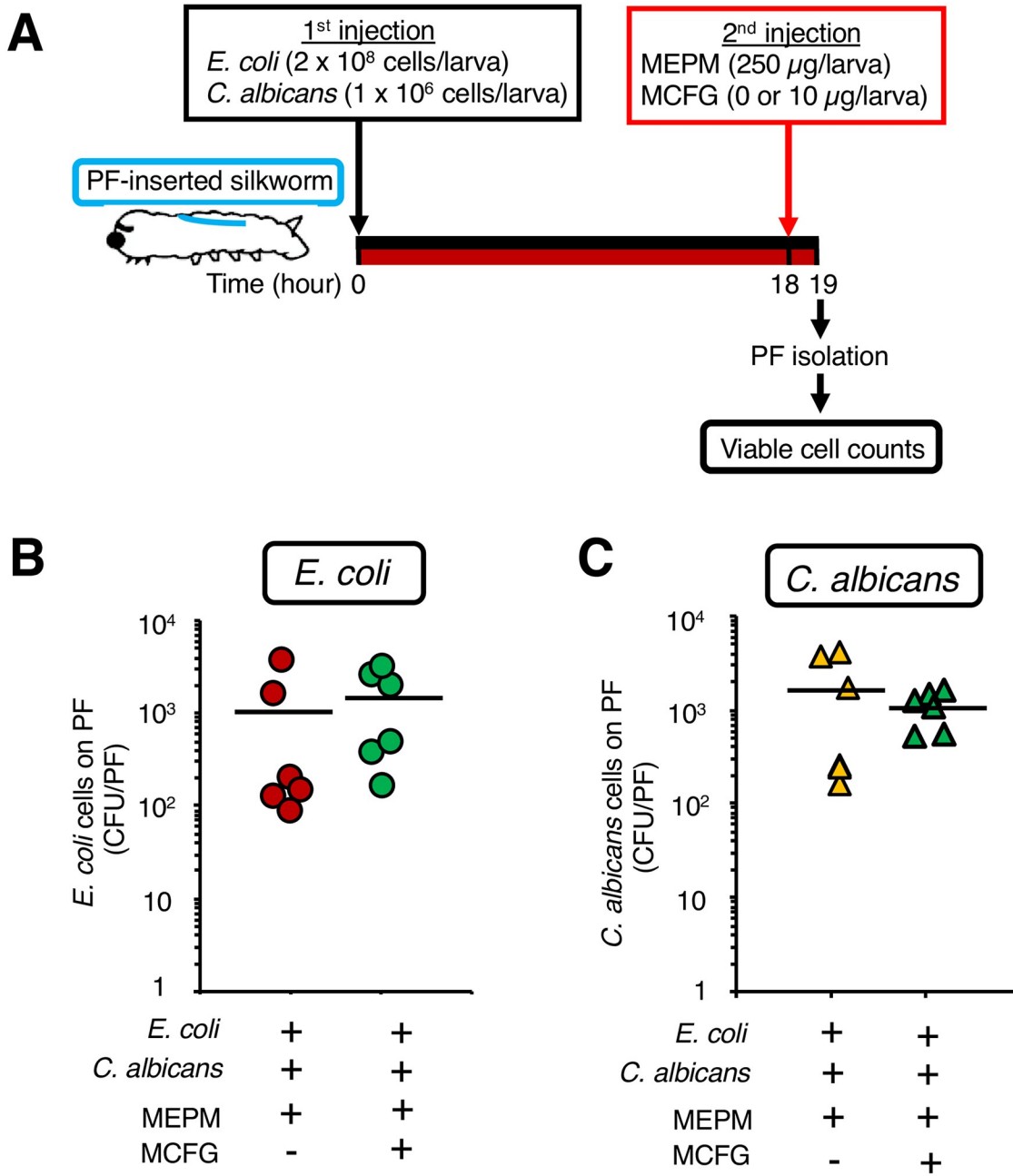

**Fig 7. Effect of MCFG on MEPM tolerance of *E. coli* induced by *C. albicans* in a dual-species biofilm on the surface of the PFs in the silkworms.** (**A**) Experimental scheme. (**B**, **C**) Mixed cell suspensions (*E. coli*: 2 x $10^8$ cells and *C. albicans*: $10^6$ cells/50 μL) were inoculated into PF-inserted silkworms, and the infected silkworms were incubated at 27˚C for 18 h. After incubation, MEPM solution (250 μg/50 μL) or a mixed drug solution (MEPM: 250 μg and MCFG: 10 μg/50 μL) were administered, and the silkworms were incubated at 27˚C for 1 h. Viable *E. coli* cells (**B**) and *C. albicans* cells (**C**) on the PF surface in silkworms were measured. n = 9/ group. Statistically significant differences between groups were evaluated using Student's *t*-test. *$P < 0.05$.

RB-3 strain exhibited sensitivity to MEPM in an *in vitro* drug susceptibility test of planktonic cells (S1 Table in S1 File) [17]. These findings suggest that clinical doses of MEPM might not be sufficient for effective treatment when a biofilm is formed by *E. coli* on the surface of a catheter *in vivo*.

Blood cultures in which *Candida* spp. are isolated are polymicrobial at 23% and dual-species biofilms with a fungal-bacterial complex are also frequently observed [15, 16, 39]. In a dual-species biofilm of the *E. coli* RB-3 and *C. albicans* SC5314 strains, the number of viable cells does not significantly decrease compared with biofilms formed by each strain *in vitro* [17]. In this study, the amounts of dual-species biofilms formed by *E. coli* RB-3 and *C. albicans* SC5314 strains in the silkworms were significantly increased compared with the amounts of single-species biofilm formed by the *E. coli* RB-3 strain. Moreover, the number of viable *E. coli* RB-3 strain cells in the biofilm did not decrease in a dual-species biofilm formed by the *E. coli* RB-3 and *C. albicans* SC5314 strains. The results suggest that *E. coli* and *C. albicans* in the silkworms form a dual-species biofilm without inhibition of the growth of either organism. Furthermore, the MEPM tolerance of the *E. coli* RB-3 strain in the dual-species biofilm with *C. albicans* in the silkworms increased compared with that in a single-species biofilm. Several *in vitro* studies have demonstrated the effect of an extracellular matrix component produced by *C. albicans* to increase the tolerance to antimicrobial agents in a dual-species biofilm [40–42]. Farnesol secreted by *C. albicans in vitro* induces gene expression in bacteria [43, 44]. No previous studies, however, have reported that *C. albicans* enhances the MEPM tolerance of *E. coli in vivo*. Our findings suggest that *C. albicans* is related to the MEPM tolerance of the *E. coli* RB-3 strain *in vivo*. Moreover, a combination of MEPM and the antifungal drug MCFG did not decrease the number of viable *E. coli* and *C. albicans* cells in a dual-species biofilm in the silkworms. The *C. albicans* SC5314 strain exhibited sensitivity to MCFG in an *in vitro* drug susceptibility test (S2 Table in S1 File). The limited efficacy of the MCFG combination in this study may be due in part to the tolerance of *C. albicans* to MCFG in a dual-species biofilm. Further studies are needed to verify whether the induction of antimicrobial tolerance in *E. coli* in a dual-species biofilm with *C. albicans in vivo* is observed with other antimicrobial agents.

Recently, techniques have been developed for high-throughput measurement of biofilm formation by *Candida* spp. using microplates [45, 46]. Based on these techniques, a method has been established to measure polymicrobial biofilm formation by *C. albicans* and *E. coli* [47]. Moreover, the method can be used to identify samples that inhibit the polymicrobial biofilm formation of *C. albicans* and *E. coli* [48]. Therefore, candidate compounds that inhibit the polymicrobial biofilm formation of *C. albicans* and *E. coli* can be obtained in future *in vitro* tests. We assumed that the effectiveness of the candidate compounds *in vivo* could be evaluated using the silkworm model developed in this study. Biofilm clarification with transparency technology has enabled the highly sensitive observation of biofilms using fluorescence microscopy [49]. In the future, the development of high-resolution observation methods that incorporate analysis using such technologies will be an important subject.

## Conclusion

An *in vivo* system of biofilm formation by *E. coli* on the surface of catheter material using silkworms was useful for evaluating the MEPM tolerance of *E. coli* and enhanced MEPM tolerance of *E. coli* caused by dual-species biofilm formation with *C. albicans*. The *in vivo* biofilm infection model using silkworms, which can be used to evaluate the effects of host factors and pharmacokinetics of antimicrobial drugs, might contribute to the development of new treatment strategies for device-associated dual-species biofilm infections.

## Supporting information

**S1 File.**
(PDF)

**S1 Dataset. Datasets included in this study.**
(XLSX)

## Acknowledgments

We thank Yu Sugiyama, Eri Sato, Sachi Koganesawa, Hiromi Kanai, Yuta Shimizu, and Mei Nakayama (Meiji Pharmaceutical University) for their technical assistance rearing the silkworms.

## Author Contributions

**Conceptualization:** Yasuhiko Matsumoto.

**Data curation:** Shintaro Eshima, Yasuhiko Matsumoto.

**Formal analysis:** Shintaro Eshima, Yasuhiko Matsumoto.

**Funding acquisition:** Takashi Sugita.

**Investigation:** Shintaro Eshima, Yasuhiko Matsumoto, Sanae Kurakado.

**Methodology:** Shintaro Eshima, Yasuhiko Matsumoto.

**Supervision:** Yasuhiko Matsumoto, Takashi Sugita.

**Validation:** Yasuhiko Matsumoto.

**Visualization:** Shintaro Eshima, Yasuhiko Matsumoto.

**Writing – original draft:** Shintaro Eshima.

**Writing – review & editing:** Yasuhiko Matsumoto, Sanae Kurakado, Takashi Sugita.

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
