## [Decision Letter · Decision Letter 0]

2 May 2023

PONE-D-23-10893Silkworm model of biofilm formation: In vivo evaluation of antimicrobial tolerance of a cross-kingdom dual-species (Escherichia coli and Candida albicans) biofilm on catheter materialPLOS ONE

Dear Dr. Matsumoto,

Thank you for submitting your manuscript to PLOS ONE. After careful consideration, we feel that it has merit but does not fully meet PLOS ONE’s publication criteria as it currently stands. Therefore, we invite you to submit a revised version of the manuscript that addresses the points raised during the review process.

We look forward to receiving your revised manuscript.

Kind regards,

Geelsu Hwang, Ph.D.

Academic Editor

PLOS ONE

Journal Requirements:

"This study was supported in part by the Research Program on Emerging and Re-Emerging Infectious Diseases of the Japan Agency for Medical Research and Development (grant number JP23fk0108679h0401 to TS)."

3. We note that Figures 1, 3, 4, 6 and 7 in your submission contain copyrighted images. All PLOS content is published under the Creative Commons Attribution License (CC BY 4.0), which means that the manuscript, images, and Supporting Information files will be freely available online, and any third party is permitted to access, download, copy, distribute, and use these materials in any way, even commercially, with proper attribution. For more information, see our copyright guidelines: http://journals.plos.org/plosone/s/licenses-and-copyright.

a. You may seek permission from the original copyright holder of Figures 1, 3, 4, 6 and 7 to publish the content specifically under the CC BY 4.0 license. 

Reviewers' comments:

Reviewer's Responses to Questions

**Comments to the Author**

1. Is the manuscript technically sound, and do the data support the conclusions?

Reviewer #1: Yes

Reviewer #2: Partly

2. Has the statistical analysis been performed appropriately and rigorously? 

Reviewer #1: Yes

Reviewer #2: Yes

3. Have the authors made all data underlying the findings in their manuscript fully available?

Reviewer #1: Yes

Reviewer #2: Yes

4. Is the manuscript presented in an intelligible fashion and written in standard English?

Reviewer #1: Yes

Reviewer #2: Yes

5. Review Comments to the Author

Reviewer #1: In the manuscript entitled “Silkworm model of biofilm formation: In vivo evaluation of antimicrobial tolerance of a cross-kingdom dual-species (Escherichia coli and Candida albicans) biofilm on catheter material”, the authors evaluate the antimicrobial tolerance of Escherichia coli and Candida albicans biofilm on catheter material. The experiments have been performed well and the manuscript is good, but requires minor revision before considering for publication in PLOS ONE. The comments for the authors are mentioned below.

1. In some places, grammar and English have to be improved. Use standard tools to correct the same.

2. Include ethical clearance certificate

3. Give justification that 1h is enough as the incubation time for MEPM. Was the experiment performed for other time durations?

4. Check the results and discussion section thoroughly. Some of the places and methods are mentioned which have to be included only in the methods section.

5. Include the following ref in the manuscript

DOI: 10.1111/jam.15402

DOI: 10.1007/s11814-021-1054-3

DOI: 10.1002/pat.5753

DOI: 10.3390/antibiotics11050573

Reviewer #2: The work has its merits especially as an in vivo silkworm model has been developed to study E. coli and its co culture with Candida albicans. However, to make the study more robust the following may be considered.

1. The authors mention that the planktonic E. coli culture was sensitive to Meropenem (MEPN) as per the previous report. What was the MIC?

Did the authors confirm the susceptibility to the antibiotic during the course of this work? If so, please include an image here. A simple agar disc diffusion experiment will clearly demonstrate the sensitivity or resistance of E. coli and C. albicans at different concentrations of the antimicrobials.

2. The silkworms have been reared on a diet containing antibiotics, please mention them. Also have you maintained a control set of silk worms that were fed a diet without antibiotics just to rule out any antimicrobial resistance conferred thereby.

3. Please justify the choice of PF as the prosthetic material in your study.

4. Please include the CFU plate images. This is crucial as one of the major claims is about the increased tolerance of E. coli to MEPN in the presence of 106 C. albicans cells as compared to 105 cells.

5. The claim about increased tolerance should be supported with broth microdilution wherein growth OD & MTT assay could be performed.

6. Please include the CFU results for concentrations below 105 and above 106 C. albicans cells.

7. Crystal violet (CV) stained Light microscopic images can be strengthened with fluorescence microscopy or SEM/CLSM. Also please mention the magnification of the bright field microscopic images.

8. As per published reports E. coli controls C. albicans growth however, as per latest reports C. albicans overcomes this inhibition at 24h. In light of this have you carried out the CFU experiment at different time points to understand the relationship dynamics in dual cultures which in turn will influence the susceptibility or resistance to drugs. Please include this perspective in your discussion as well.

9. In vitro models for optimization of growth of Candida and E. coli individual and cocultures have been developed on several prosthetic materials. Have you optimized the conditions for growth of the culture on polyurethane foam?

10. Please compare the merits of your in vivo model with that of published in vitro models.

6. PLOS authors have the option to publish the peer review history of their article (what does this mean?). If published, this will include your full peer review and any attached files.

Reviewer #1: No

Reviewer #2: **Yes: **Bindu S

---

## [Author Response · Author response to Decision Letter 0]

31 May 2023

According to the editor’s comment, we revised the manuscript based on the PLOS ONE style templates.

"This study was supported in part by the Research Program on Emerging and Re-Emerging Infectious Diseases of the Japan Agency for Medical Research and Development (grant number JP23fk0108679h0401 to TS)."

According to the editor’s comment, the Funding Statement was included in the cover letter. 

This study was supported in part by the Research Program on Emerging and Re-Emerging Infectious Diseases of the Japan Agency for Medical Research and Development (grant number JP23fk0108679h0401 to TS) and for Scientific Research (C) to YM (JP23K06141). The funders had no role in the study design, data collection, data analysis, decision to publish, or preparation of the manuscript. There was no additional external funding received for this study.

3. We note that Figures 1, 3, 4, 6 and 7 in your submission contain copyrighted images. All PLOS content is published under the Creative Commons Attribution License (CC BY 4.0), which means that the manuscript, images, and Supporting Information files will be freely available online, and any third party is permitted to access, download, copy, distribute, and use these materials in any way, even commercially, with proper attribution. For more information, see our copyright guidelines: http://journals.plos.org/plosone/s/licenses-and-copyright.

a. You may seek permission from the original copyright holder of Figures 1, 3, 4, 6 and 7 to publish the content specifically under the CC BY 4.0 license. 

The image of silkworm was produced by me and firstly published in PLoS One (Matsumoto Y., et al., PLoS One, 2011). According to the editor’s comment, we added the Content Permission Form.

[Reference]

Matsumoto Y, Sumiya E, Sugita T, Sekimizu K. An invertebrate hyperglycemic model for the identification of anti-diabetic drugs. PLoS One. 2011 6:e18292. doi: 10.1371/journal.pone.0018292.

Reviewer #1: In the manuscript entitled “Silkworm model of biofilm formation: In vivo evaluation of antimicrobial tolerance of a cross-kingdom dual-species (Escherichia coli and Candida albicans) biofilm on catheter material”, the authors evaluate the antimicrobial tolerance of Escherichia coli and Candida albicans biofilm on catheter material. The experiments have been performed well and the manuscript is good, but requires minor revision before considering for publication in PLOS ONE. The comments for the authors are mentioned below.

1. In some places, grammar and English have to be improved. Use standard tools to correct the same.

Following the referee’s comment, the revised manuscript was edited by professional native English-speaking science editors (SciTechEdit International, LLC, CO, USA).

2. Include ethical clearance certificate

Following the referee’s comment, we added the file of the ethical clearance certificate.

3. Give justification that 1h is enough as the incubation time for MEPM. Was the experiment performed for other time durations?

In an experiment with silkworms 6 h after the administration of MEPM, some silkworms died in the saline group as the control. In addition, the half-life of MEPM in humans after reaching its maximum blood concentration is approximately 1 (Johan W., et al. Clinical Pharmacokinetics, 1995). Therefore, we examined one hour after inoculation for evaluating the effects of MEPM.

[Reference]

Johan W. Mouton & John N. van den Anker. Meropenem Clinical Pharmacokinetics. Clinical Pharmacokinetics. 1995 28:275–286.　

4. Check the results and discussion section thoroughly. Some of the places and methods are mentioned which have to be included only in the methods section.

According to the reviewer’s comment, we changed the places of the sentences appropriately in the revised manuscript. 

Moreover, the following sentences have been removed from the Results section of the revised manuscript.

・These results suggest E. coli (2 × 108 cells/larva) inoculated into PF-inserted silkworms formed a biofilm on the PF surface in the silkworms.

・These results suggest that the dual-species biofilm formed by E. coli and C. albicans on the PF surface in the silkworms does not inhibit the growth of either organism.

・These results suggest that C. albicans in a dual-species biofilm with E. coli promotes the MEPM tolerance of E. coli in silkworms.

・These results suggest that MEPM tolerance of E. coli in a dual-species biofilm are not suppressed by co-administration with the antifungal drug MCFG.

5. Include the following ref in the manuscript

DOI: 10.1111/jam.15402

DOI: 10.1007/s11814-021-1054-3

DOI: 10.1002/pat.5753

DOI: 10.3390/antibiotics11050573

According to the reviewer’s comment, we added the references in the Discussion section of the revised manuscript (Page 20, lines 334-341).

[Page 20, lines 334-341]

Recently, techniques have been developed for high-throughput measurement of biofilm formation by Candida spp. using microplates (Sadanandan B., et al., Journal of Applied Microbiology, 2021, Sadanandan B., et al., Korean J. Chem Eng., 2022). Based on these techniques, a method has been established to measure polymicrobial biofilm formation by C. albicans and E. coli (Ashrit P., et al., Polymers Advanced Technologies, 2022). Moreover, the method can be used to identify samples that inhibit the polymicrobial biofilm formation of C. albicans and E. coli (Ashrit P., et al., Antibiotics, 2022). Therefore, candidate compounds that inhibit the polymicrobial biofilm formation of C. albicans and E. coli can be obtained in future in vitro tests. We assumed that the effectiveness of the candidate compounds in vivo could be evaluated using the silkworm model developed in this study.

[References]

・Sadanandan B, Vaniyamparambath V, Lokesh KN, Shetty K, Joglekar AP, Ashrit P, Hemanth B. Candida albicans biofilm formation and growth optimization for functional studies using response surface methodology. J Appl Microbiol. 2022 132:3277-3292. doi: 10.1111/jam.15402.

・Sadanandan B, Ashrit P, Nataraj LK, Shetty K, Jogalekar AP, Vaniyamparambath V, Hemanth B. High throughput comparative assessment of biofilm formation of Candida glabrata on polystyrene material. Korean J Chem Eng. 2022 39:1277-1286. 

・Ashrit P, Sadanandan B, Nataraj LK, Shetty K, Vaniyamparambath V, Raghu AV. A microplate-based Response Surface Methodology model for growth optimization and biofilm formation on polystyrene polymeric material in a Candida albicans and Escherichia coli co-culture. Polymers for Advanced Technologies. 2022 33:2872-2885.

・Ashrit P, Sadanandan B, Shetty K, Vaniyamparambath V. Polymicrobial Biofilm Dynamics of Multidrug-Resistant Candida albicans and Ampicillin-Resistant Escherichia coli and Antimicrobial Inhibition by Aqueous Garlic Extract. Antibiotics (Basel). 2022 11:573. doi: 10.3390/antibiotics11050573.

Reviewer #2: The work has its merits especially as an in vivo silkworm model has been developed to study E. coli and its co culture with Candida albicans. However, to make the study more robust the following may be considered.

1. The authors mention that the planktonic E. coli culture was sensitive to Meropenem (MEPN) as per the previous report. What was the MIC?

Did the authors confirm the susceptibility to the antibiotic during the course of this work? If so, please include an image here. A simple agar disc diffusion experiment will clearly demonstrate the sensitivity or resistance of E. coli and C. albicans at different concentrations of the antimicrobials.

According to the reviewer’s comment, we determined the MIC value of MEPM against E. coli RB-3. Susceptibility testing for meropenem (MEPM) was performed using the MicroScan AST panel (Beckman Coulter, Pasadena, CA, USA) according to CLSI M100-Ed31. The MIC value of MEPM was less than 1 µg/mL. The result was described in Supplementary Table 1 (S1 Table) of the revised manuscript (Page 19, line 308). Because the fungus C. albicans is not sensitive to the antibacterial drug MEPM, observation of the inhibitory circle by the disc diffusion experiment is difficult on plates with C. albicans and E. coli.

[Reference]

Clinical and Laboratory Standards Institute (CLSI). Performance standards for antimicrobial susceptibility testing; M100-Ed30. 30th ed. Wayne, PA: CLSI; 2020.

2. The silkworms have been reared on a diet containing antibiotics, please mention them. Also have you maintained a control set of silk worms that were fed a diet without antibiotics just to rule out any antimicrobial resistance conferred thereby.

Following the reviewer’s comment, we added the description that the silkworms reared on a diet containing antibiotics in the Materials & Methods section of the revised manuscript (Page 6, lines 99-100). The artificial diet without antibiotics was fed 24 hours before the infection experiments (Page 6, lines 100-101).

[Page 6, lines 99-101]

Silkworms were fed an artificial diet, Silkmate 2S, containing antibiotics purchased from Ehime-Sanshu Co., Ltd. (Ehime, Japan). Fifth-instar silkworms were fed with an antibiotic-free artificial diet (Sysmex Corporation, Hyogo, Japan) for 1 day.

3. Please justify the choice of PF as the prosthetic material in your study.

According to the reviewer’s comment, we changed the sentence in the revised manuscript (Page 5, lines 74-77). 

[Page 5, lines 74-77]

In this study, we established an experimental system for evaluating biofilm formation by E. coli on the surface of a polyurethane fiber (PF) inserted in a silkworm and investigated the effect of MEPM against a dual-species biofilm with C. albicans. Polyurethane is a central venous catheter material (Ju DB., et al., Tissue Eng Regen Med. 2022).

[Reference]

Ju DB, Lee JC, Hwang SK, Cho CS, Kim HJ. Progress of Polysaccharide-Contained Polyurethanes for Biomedical Applications. Tissue Eng Regen Med. 2022 19(5):891-912. doi: 10.1007/s13770-022-00464-2.

4. Please include the CFU plate images. This is crucial as one of the major claims is about the increased tolerance of E. coli to MEPN in the presence of 106 C. albicans cells as compared to 105 cells.

According to the reviewer’s comment, we added the plate images in the Supplementary Fig. 1 (S1 Fig) of the revised manuscript (Page 17, line 281). In this study, viable E. coli cells on the surface of the PFs in the silkworms were grown on nutrient agar medium containing micafungin (1 μg/mL), which inhibits the growth of C. albicans. The sentences were described in the Materials & Methods section of the revised manuscript (Page 8, lines 136-139).

[Page 8, lines 136-139]

The CFUs of E. coli and C. albicans were calculated: 1 μg/mL micafungin was added to nutrient agar medium (to selectively grow only E. coli) or 100 μg/mL streptomycin was added to Sabouraud agar medium (to selectively grow only C. albicans). E. coli RB-3 and C. albicans SC5314 strains were susceptible to streptomycin and micafungin, respectively.

5. The claim about increased tolerance should be supported with broth microdilution wherein growth OD & MTT assay could be performed.

Because OD and MTT assays cannot distinguish between the survival of E. coli and C. albicans, we assessed each by measuring their respective colony counts in this study. We used media selective for each organism, such as media containing antifungal drug micafungin where E. coli proliferates selectively, and media containing antibacterial drug streptomycin where C. albicans grows selectively, to measure the colony counts. We added the sentences to clarify this point in the Materials & Methods section of the revised manuscript (Page 8, lines 136-139).

[Page 8, lines 136-139]

The CFUs of E. coli and C. albicans were calculated: 1 μg/mL micafungin was added to nutrient agar medium (to selectively grow only E. coli) or 100 μg/mL streptomycin was added to Sabouraud agar medium (to selectively grow only C. albicans). E. coli RB-3 and C. albicans SC5314 strains were susceptible to streptomycin and micafungin, respectively.

6. Please include the CFU results for concentrations below 105 and above 106 C. albicans cells.

We performed the dose response of C. albicans cells on the tolerance to MEPM of E.coli. The addition of C. albicans (104-105 cells) did not increase the tolerance of MEPM of E. coli. The result is shown in Fig. 6C of the revised manuscript.

7. Crystal violet (CV) stained Light microscopic images can be strengthened with fluorescence microscopy or SEM/CLSM. Also please mention the magnification of the bright field microscopic images.

According to the reviewer’s comment, we added the below sentences in the revised manuscript (Pages 20-21, lines 341-344). Moreover, we added the magnification of the bright field microscopic images on the Fig. 1B and 4B in the revised manuscript.

[Pages 20-21, lines 341-344]

Biofilm clarification with transparency technology has enabled the highly sensitive observation of biofilms using fluorescence microscopy (Sugimoto S et al., Communications Biology, 2023). In the future, the development of high-resolution observation methods that incorporate analysis using such technologies will be an important subject.

[Reference]

Sugimoto S, Kinjo Y. Instantaneous Clearing of Biofilm (iCBiofilm): an optical approach to revisit bacterial and fungal biofilm imaging.

Commun Biol. 2023 6:38. doi: 10.1038/s42003-022-04396-4.

8. As per published reports E. coli controls C. albicans growth however, as per latest reports C. albicans overcomes this inhibition at 24h. In light of this have you carried out the CFU experiment at different time points to understand the relationship dynamics in dual cultures which in turn will influence the susceptibility or resistance to drugs. Please include this perspective in your discussion as well.

According to the reviewer’s comment, we added the sentences in the Discussion section of the revised manuscript (Pages 17-18, lines 288-291). We assumed that E. coli affects the growth of C. albicans at the early time point.

[Pages 17-18, lines 288-291]

In this study, we investigated biofilm formation at 18 h after inoculation with E. coli and C. albicans. In an in vitro experiment, E. coli inhibits the growth of C. albicans (Cabral DJ et. al., Microb Cell, 2018). We assumed that prolonged co-infection in the host environment was necessary for the formation of a dual-species biofilm for tolerance to MEPM.

[Reference]

Cabral DJ, Penumutchu S, Norris C, Morones-Ramirez JR, Belenky P. Microbial competition between Escherichia coli and Candida albicans reveals a soluble fungicidal factor. Microb Cell. 2018 5:249-255. doi: 10.15698/mic2018.05.631.

9. In vitro models for optimization of growth of Candida and E. coli individual and cocultures have been developed on several prosthetic materials. Have you optimized the conditions for growth of the culture on polyurethane foam?

We optimized the conditions for C. albicans SC5314 strain to form biofilms on the surface of polyurethane fibers in vitro and in vivo using silkworms (Matsumoto Y., et al., Med. Mycol., 2021). Furthermore, we optimized biofilm formation by co-cultivation of the C. albicans SC5314 and E. coli RB-3 strains used in this study in vitro (Eshima S., et al., Microorganisms, 2022). Based on the information, we further optimized the amount of E. coli or C. albicans administered to silkworms in this study.

[References]

・Matsumoto Y, Kurakado S, Sugita T. Evaluating Candida albicans biofilm formation in silkworms.

Med Mycol. 2021 59:201-205. doi: 10.1093/mmy/myaa064.

・Eshima S, Kurakado S, Matsumoto Y, Kudo T, Sugita T. Candida albicans Promotes the Antimicrobial Tolerance of Escherichia coli in a Cross-Kingdom Dual-Species Biofilm.

Microorganisms. 2022 10:2179. doi: 10.3390/microorganisms10112179.

10. Please compare the merits of your in vivo model with that of published in vitro models.

According to the reviewer’s comment, we added the sentences in the Discussion section of the revised manuscript (Page 21, lines 349-352). 

[Page 21, lines 349-352]

The in vivo biofilm infection model using silkworms, which can be used to evaluate the effects of host factors and pharmacokinetics of antimicrobial drugs, might contribute to the development of new treatment strategies for device-associated dual-species biofilm infections.

---

## [Decision Letter · Decision Letter 1]

19 Jun 2023

PONE-D-23-10893R1Silkworm model of biofilm formation: In vivo evaluation of antimicrobial tolerance of a cross-kingdom dual-species (Escherichia coli and Candida albicans) biofilm on catheter materialPLOS ONE

Dear Dr. Matsumoto,

Thank you for submitting your manuscript to PLOS ONE. After careful consideration, we feel that it has merit but does not fully meet PLOS ONE’s publication criteria as it currently stands. Therefore, we invite you to submit a revised version of the manuscript that addresses the points raised during the review process.

We look forward to receiving your revised manuscript.

Kind regards,

Geelsu Hwang, Ph.D.

Academic Editor

PLOS ONE

Journal Requirements:

Additional Editor Comments (if provided):

One of the reviewers still insists to include MIC data to be accepted for publication. Please also address other comments raised by the reviewer.

Reviewers' comments:

Reviewer's Responses to Questions

**Comments to the Author**

1. If the authors have adequately addressed your comments raised in a previous round of review and you feel that this manuscript is now acceptable for publication, you may indicate that here to bypass the “Comments to the Author” section, enter your conflict of interest statement in the “Confidential to Editor” section, and submit your "Accept" recommendation.

Reviewer #1: All comments have been addressed

Reviewer #2: (No Response)

2. Is the manuscript technically sound, and do the data support the conclusions?

Reviewer #1: Yes

Reviewer #2: Partly

3. Has the statistical analysis been performed appropriately and rigorously? 

Reviewer #1: Yes

Reviewer #2: Yes

4. Have the authors made all data underlying the findings in their manuscript fully available?

Reviewer #1: Yes

Reviewer #2: Yes

5. Is the manuscript presented in an intelligible fashion and written in standard English?

Reviewer #1: Yes

Reviewer #2: Yes

6. Review Comments to the Author

Reviewer #1: The manuscript entitled “Silkworm model of biofilm formation: In vivo evaluation of antimicrobial tolerance of a cross-kingdom dual-species (Escherichia coli and Candida albicans) biofilm on catheter material”, the authors evaluate the antimicrobial tolerance of Escherichia coli and Candida albicans biofilm on catheter material. The experiments have been performed well and the manuscript is good

Reviewer #2: Review Comments

1. The Microscan AST in place of agar disc diffusion carried out for E. coli is acceptable.

Candida albicans sensitivity or resistance to the antimicrobial was meant for the relevant antimicrobial micafungin & not Meropenem (MEPM), therefore the term antimicrobials had been used, so please include the MIC information for Candida as well.

2. The antibiotics present in the silkworm artificial diet have not been included so please include them.

3. Experiments on Growth OD & MTT were suggested as in the earlier manuscript version MIC was not included. Now that Microscan AST has been done to determine MIC of MEPM against E. coli it is OK. However as indicated in point 1 the MIC information for Candida should also be included.

7. PLOS authors have the option to publish the peer review history of their article (what does this mean?). If published, this will include your full peer review and any attached files.

Reviewer #1: No

Reviewer #2: No

---

## [Author Response · Author response to Decision Letter 1]

24 Jun 2023

Reviewer #1: The manuscript entitled “Silkworm model of biofilm formation: In vivo evaluation of antimicrobial tolerance of a cross-kingdom dual-species (Escherichia coli and Candida albicans) biofilm on catheter material”, the authors evaluate the antimicrobial tolerance of Escherichia coli and Candida albicans biofilm on catheter material. The experiments have been performed well and the manuscript is good.

Thank you very much for the time spent reviewing our manuscript. 

Reviewer #2: Review Comments

1. The Microscan AST in place of agar disc diffusion carried out for E. coli is acceptable.

Candida albicans sensitivity or resistance to the antimicrobial was meant for the relevant antimicrobial micafungin & not Meropenem (MEPM), therefore the term antimicrobials had been used, so please include the MIC information for Candida as well.

According to the reviewer’s comment, we determined MIC value of micafungin against Candida albicans SC5314. The result was described in Supplementary Table S2. Furthermore, we described the below sentence in the Discussion section of the revised manuscript (Page 20, lines 329-330).

[Page 20, lines 329-330]

The C. albicans SC5314 strain exhibited sensitivity to MCFG in an in vitro drug susceptibility test (S2 Table in S1 File).

2. The antibiotics present in the silkworm artificial diet have not been included so please include them.

According to the reviewer’s comment, we described the below sentence in the Materials and Methods section of the revised manuscript (Page 6, lines 99-100). Since vancomycin is not absorbed from intestinal tract to hemolymph, we assumed that diet containing vancomycin does not affect the infection experiments with E. coli. 

[Page 6, lines 99-100]

Silkworms were fed an artificial diet, Silkmate 2S (Ehime-Sanshu Co., Ltd., Ehime, Japan) mixed with vancomycin (300 µg/g of Silkmate 2S).

3. Experiments on Growth OD & MTT were suggested as in the earlier manuscript version MIC was not included. Now that Microscan AST has been done to determine MIC of MEPM against E. coli it is OK. However as indicated in point 1 the MIC information for Candida should also be included.

According to the reviewer’s comment, we determined MIC value of micafungin against Candida albicans SC5314. The result was described in Supplementary Table S2. Furthermore, we described the below sentence in the Discussion section of the revised manuscript (Page 20, lines 329-330).

[Page 20, lines 329-330]

The C. albicans SC5314 strain exhibited sensitivity to MCFG in an in vitro drug susceptibility test (S2 Table in S1 File).

Thank you very much for the time spent reviewing our manuscript.

---

## [Decision Letter · Decision Letter 2]

28 Jun 2023

Silkworm model of biofilm formation: In vivo evaluation of antimicrobial tolerance of a cross-kingdom dual-species (Escherichia coli and Candida albicans) biofilm on catheter material

PONE-D-23-10893R2

Dear Dr. Matsumoto,

We’re pleased to inform you that your manuscript has been judged scientifically suitable for publication and will be formally accepted for publication once it meets all outstanding technical requirements.

Kind regards,

Geelsu Hwang, Ph.D.

Academic Editor

PLOS ONE

Reviewers' comments:

Reviewer's Responses to Questions

**Comments to the Author**

1. If the authors have adequately addressed your comments raised in a previous round of review and you feel that this manuscript is now acceptable for publication, you may indicate that here to bypass the “Comments to the Author” section, enter your conflict of interest statement in the “Confidential to Editor” section, and submit your "Accept" recommendation.

Reviewer #2: All comments have been addressed

2. Is the manuscript technically sound, and do the data support the conclusions?

Reviewer #2: Yes

3. Has the statistical analysis been performed appropriately and rigorously? 

Reviewer #2: Yes

4. Have the authors made all data underlying the findings in their manuscript fully available?

Reviewer #2: Yes

5. Is the manuscript presented in an intelligible fashion and written in standard English?

Reviewer #2: Yes

6. Review Comments to the Author

Reviewer #2: The authors have satisfactorily addressed all the review comments. Therefore the manuscript is recommended for publication.

7. PLOS authors have the option to publish the peer review history of their article (what does this mean?). If published, this will include your full peer review and any attached files.

Reviewer #2: No

---

## [Editor Report · Acceptance letter]

5 Jul 2023

PONE-D-23-10893R2 

Silkworm model of biofilm formation: In vivo evaluation of antimicrobial tolerance of a cross-kingdom dual-species (*Escherichia coli* and *Candida albicans*) biofilm on catheter material 

Dear Dr. Matsumoto:

I'm pleased to inform you that your manuscript has been deemed suitable for publication in PLOS ONE. Congratulations! Your manuscript is now with our production department. 

Kind regards, 

on behalf of

Dr. Geelsu Hwang 

Academic Editor

PLOS ONE